# Structural Characterization of a *Polygonatum cyrtonema* Hua Tuber Polysaccharide and Its Contribution to Moisture Retention and Moisture-Proofing of Porous Carbohydrate Material

**DOI:** 10.3390/molecules27155015

**Published:** 2022-08-06

**Authors:** Ling Yu, Yipeng Wang, Qingjiu Tang, Rongrong Zhang, Danyu Zhang, Guangyong Zhu

**Affiliations:** 1Department of Perfume and Aroma Technology, Shanghai Institute of Technology, No. 100 Haiquan Road, Shanghai 201418, China; 2Institute of Edible Fungi, Shanghai Academy of Agricultural Sciences, Shanghai 201403, China

**Keywords:** *Polygonatum cyrtonema* Hua polysaccharide, structural characterization, moisture retention and moisture-proofing

## Abstract

Porous carbohydrate materials such as tobacco shreds readily absorb moisture and become damp during processing, storage, and consumption (smoking). Traditional humectants have the ability of moisture retention but moisture-proofing is poor. *Polygonatum cyrtonema* Hua polysaccharide (PCP 85−1−1) was separated by fractional precipitation and was purified by anion exchange and gel permeation chromatography. The average molecular weight (Mw) of PCP 85−1−1 was 2.88 × 10^3^ Da. The monosaccharide composition implied that PCP 85−1−1 consisted of fucose, glucose, and fructose, and the molar ratio was 22.73:33.63:43.65. When 2% PCP 85−1−1 was added to tobacco shreds, the ability of moisture retention and moisture-proofing were significantly enhanced. The moisture retention index (MRI) and moisture-proofing index (MPI) increased from 1.95 and 1.67 to 2.11 and 2.14, respectively. Additionally, the effects of PCP 85−1−1 on the aroma and taste of tobacco shreds were evaluated by electronic tongue and gas chromatography–mass spectrometry (GC-MS). These results indicated that PCP 85−1−1 had the characteristics of preventing water absorption under high relative humidity and moisturizing under dry conditions. The problem that traditional humectants are poorly moisture-proof was solved. PCP 85−1−1 can be utilized as a natural humectant on porous carbohydrates, which provides a reference for its development and utilization.

## 1. Introduction

The structure of tobacco shreds contains many capillaries, hydrophilic colloids, and crystalline components, which are representative of porous carbohydrate materials. It is susceptible to environmental conditions such as temperature and relative humidity (RH). Polyol compounds, such as glycerol, propylene glycol, and sorbitol (traditional humectants), have been widely used in the tobacco industry to improve the moisture retention of tobacco shreds [1,2,3]. The moisture retention activities of these compounds are derived from their hydroxyl groups, which form hydrogen bonds with water molecules. However, these traditional humectants have strong moisture absorption characteristics that may alter the moisture content of tobacco shreds and potentially affect the taste of tobacco during burning and smoking. Additionally, these compounds can release propylene oxide, acrolein, and other compounds during tobacco smoking, contributing to adverse health effects. Therefore, researchers have been looking for a humectant that not only has the function of moisture retention and moisture-proofing but can also improve the feeling of tobacco smoking [4,5].

Polysaccharides are generally strong hydrophilic compounds and possess excellent moisture absorption and retention capacities. Polysaccharides and natural extract humectants generally have good hydrophilic properties, thus providing new ideas and directions for improving the water-holding capacity of cigarettes [3]. However, most of these humectants showed a moisturizing effect better than traditional humectants, but the improvement of the moisture-proofing effect of cigarettes is not very obvious. For example, Chen et al. [6] introduced a hydrophilic group into D-mannose to prepare 1-O-carboxymethyl-D-mannofuranoose, which had better moisturizing properties than propylene glycol and glycerin, but its moisture-proofing activities were worse under high relative humidity.

In China, plants of the *Polygonatum* genus, including *Polygonatum sibiricum* Red, *Polygonatum cyrtonema* Hua, and *Polygonatum kingianum* Coll. et Hemsl. are widely used for medicine and food [7]. *Polygonatum cyrtonema* Hua is distributed throughout the temperate Northern Hemisphere in places such as China, Japan, Korea, India, Russia, Europe, and North America [8]. At present, it has been planted as an interplanting understory crop in the Sanming area, Fujian Province, China [9,10,11]. *Polygonatum cyrtonema* Hua contains multiple components, such as alkaloids, flavones, steroid saponins, lignins, amino acids, and *Polygonatum cyrtonema* Hua polysaccharides (PCP) [12].

*Polygonatum cyrtonema* Hua polysaccharides are believed to be one of the most important active compounds of *Polygonatum cyrtonema* Hua. Studies have shown that they have an antioxidative [13], anti-bacterial [14] and anti-tumor [15] ability. Additionally, they also can promote the reduction of blood sugar and blood lipids [16], improve the immune system [17,18], and possess moisture retention and moisture-proofing properties [19]. According to the previous research results [20,21,22], fractional precipitation with ethanol has been commonly used as a convenient and rapid method for the initial purification of water-soluble polysaccharides. Importantly, the required concentration of ethanol is related to the molecular size, structure, and biological activity of polysaccharides. The structures of polysaccharides obtained by different purification methods were different. Interestingly, previous research has shown that polysaccharides from *Polygonatum*
*cyrtonema* Hua were mainly fructooligosaccharides (FOS) with fructose as the structural backbone. He et al. [23] found two fructooligosaccharides (PFOS) that were graminan-type fructan with a degree of polymerization ranging from 5 to 10. Zhang et al. [24] also obtained oligosaccharides from *Polygonatum*
*cyrtonema* Hua. Structural analysis suggested that the fructan consisted of a (2 → 6) linked β-D-Fruf residues backbone with an internal α-D-Glcp residue and two (2 → 1) linked β-D-Fruf residue branches. Therefore, in this study, PCP 85−1−1 was purified by graded alcohol precipitation, and its physicochemical properties, moisture retention, and moisture-proofing abilities were further studied.

However, the reports on the moisture retention and moisture-proofing functions of *Polygonatum cyrtonema* Hua polysaccharides on porous carbohydrate materials are scarce. In this study, we investigated the influence of *Polygonatum cyrtonema* Hua polysaccharides on the adsorption and desorption properties of tobacco shreds. Structural characteristics of *Polygonatum cyrtonema* Hua polysaccharides also were analyzed. Additionally, the effect of the *Polygonatum cyrtonema* Hua polysaccharides on the changes in taste and aroma components of tobacco shreds after smoking were analyzed by electronic tongue and gas chromatography–mass spectrometry (GC-MS).

## 2. Results and Discussion

### 2.1. Extraction and Comparison of Ethanol Precipitated Fractions from Water Extracts

The yield of *Polygonatum cyrtonema* Hua polysaccharides by fractional precipitation is shown in Table 1. The yield of PCP 85 was the highest, suggesting *Polygonatum cyrtonema* Hua polysaccharides were mainly low molecular polysaccharides. The purity of PCP 20 was the highest, at 62.44 ± 1.64% ^d^. As shown in Table 1, the yields of PCP 20, PCP 40, PCP 60, and PCP 85 were 1.34 ± 0.88% ^a^, 5.32 ± 0.54% ^c^, 2.47 ± 0.61% ^b^, and 15.43 ± 1.21% ^d^, respectively (*p* < 0.05); the purity was 62.44 ± 1.64% ^d^, 56.64 ± 3.57% ^a^, 57.77 ± 1.58% ^b^, and 58.18 ± 2.25% ^c^, respectively; the contents of the protein were 3.98 ± 0.0075 ^c^ mg/g, 3.15 ± 0.10 ^a^ mg/g, 3.83 ± 0.015 ^c^ mg/g, and 3.45 ± 0.11 ^b^ mg/g, respectively; and the polyphenol contents were 5.54 ± 0.21 ^d^ mg/g, 2.76 ± 0.059 ^c^ mg/g, 2.28 ± 0.036 ^b^ mg/g, and 1.61 ± 0.001 ^a^ mg/g, respectively (*p* < 0.05). The yield of PCP 85 was different from a previous study finding that Zhang et al. [24] obtained with DPC1 with yields of 11.40%. The differences in the yield of PCP 85 may be that Zhang et al. obtained DPC1 with an ethanol concentration of 80% (*v*/*v*), while PCP 85 was obtained with an ethanol concentration of 85% (*v*/*v*) in this study.

### 2.2. Moisture Absorption and Moisture Retention Properties of PCP

#### 2.2.1. Moisture Absorption Properties

The moisture absorption values of *Polygonatum cyrtonema* Hua polysaccharides, hyaluronic acid, and sorbitol at RH 81% are shown in Figure 1A. The moisture absorption of each sample was increased sharply within 0–24 h. The change rate gradually decreased, and the curve tended to be flat with the increase in time. The moisture absorption rates were almost saturated after 24 h. At this time, the moisture absorption rates of hyaluronic acid, sorbitol, PCP 20, PCP 40, PCP 60, and PCP 85 were 33.27 ± 0.84% ^f^, 3.38 ± 1.02% ^a^, 24.96 ± 1.06% ^d^, 26.60 ± 2.33% ^e^, 19.43 ± 1.08% ^b^, and 21.41 ± 0.31% ^c^, respectively (*^a–f^ indicates a significantly different from each other (analysis of variance, p < 0.05)*). Comparative moisture absorption properties of the polysaccharides, hyaluronic acid and sorbitol were as follows: hyaluronic acid > PCP 40 > PCP 20 > PCP 85 > PCP 60 > sorbitol (*p* < 0.05). The results showed that PCP 20 and PCP 40 had the best moisture absorption properties, whereas PCP 60 and PCP 85 had the best moisture-proofing properties. This results of PCP 40 (26.60%) and PCP 85 (24.14%) are better than Chou [25], who found that polysaccharides (PNP-40, PNP-60, PNP-80) from *Pholiota nameko* by fractional precipitation showed PNP-40 (14.37%) and PNP-80 (18.00%) moisture absorption at 81% RH after 24 h. However, PCP 60 (19.43%) was lower than PNP-60 (21.48%). The content of uronic acid in PNP-60 was speculated to be the reason for its better moisture absorption ability. Although hyaluronic acid has an efficient moisturizing effect, compared with hyaluronic acid, natural polysaccharides have moisture absorption properties as well as a variety of physiological activities, such as anti-oxidant activity [13]. The application of polysaccharides in porous carbohydrates (tobacco shreds) also has good potential for development.

#### 2.2.2. Moisture Retention Properties

As shown in Figure 1B, the change in the curves showed that the moisture retention of each sample decreased sharply in the first 12 h. The curve tended to be flat with the increase of time. After 12 h, the moisture retention of each sample had been maintained in a stable state. The moisture retention rates of hyaluronic acid, sorbitol, PCP 20, PCP 40, PCP 60, and PCP 85 were 85.58 ± 0.81% ^b^, 80.10 ± 0.99% ^a^, 85.59 ± 0.85% ^b^, 85.37 ± 0.78% ^b^, 88.75 ± 1.14% ^d^, and 86.59 ± 1.01% ^c^, respectively (*^a–d^ indicates a significantly different from each other (analysis of variance, p < 0.05)*). Comparative moisture retention properties of polysaccharides, hyaluronic acid, and sorbitol were as follows: PCP 60 > PCP 85 > PCP 20 > hyaluronic acid > PCP 40 > sorbitol. Among the polysaccharides, PCP 60 had the best moisture retention (*p* < 0.05). Hyaluronic acid is a common humectant that shows good moisture retention performance [26]. It has been reported that the moisture retention rate of enzymatic *Sargassum*
*horneri* polysaccharides in dry silica gel after 48 h is about 53% [27]. Together, these results demonstrate the strong moisture retention capacity of PCP polysaccharides, as compared to hyaluronic acid and other plant polysaccharides. The major reason for the moisture retention ability may be that the polar group in the polysaccharide can form a hydrogen bond with H_2_O to unify the massive moisture contents. The chain of polysaccharides can also mutually interweave with the lattice, which plays a very strong role in guaranteeing moisture content [28].

The above results show that PCP 60 and PCP 85 had the characteristics of preventing water absorption under high relative humidity and moisturizing under dry conditions. In conclusion, *Polygonatum cyrtonema* Hua polysaccharides can be utilized as a natural humectant on porous carbohydrates (tobacco shreds).

### 2.3. Purification, Chemical, and Conformational Characteristics of PCP 85−1−1

Based on the yield and the moisture absorption and moisture retention activity, the fraction (PCP 85) with the best activity was selected and further purified through anion exchange and gel permeation chromatography to characterize the structure. PCP 85 was initially separated in a DEAE–Sepharose Fast Flow cellulose column (Figure 2A). The fractions eluted with deionized water were of a much higher content than the other fractions from the crude polysaccharides (PCP 85). Then the deionized water fractions were further purified with a Superdex 75 pg column (Figure 2B). Purified polysaccharides from *Polygonatum cyrtonema* Hua (PCP85-1-1) were obtained.

As shown in Figure 3A, PCP 85−1−1 displayed a single symmetrical peak, indicating it is a homogeneous polysaccharide. Based on its retention time on the HPGPC system, the average molecular weight of PCP 85−1−1 was calculated to be 2.88 kDa. Similar results were obtained in other reports [29,30], which obtained *Polygonatum cyrtonema* Hua polysaccharides with a molecular weight of 2–5 kDa. These studies showed PCP was a low-weight natural polysaccharide. However, in these reports, *Polygonatum cyrtonema* Hua polysaccharides were a fructan. The monosaccharides of PCP 85 were different from the previous study, which were determined by HPLC-RID analysis. As shown in Figure 3B, PCP 85−1−1 was composed of fucose, glucose, and fructose, and the molar ratio was 22.73:33.63:43.65. The differences in the monosaccharides may depend on the sample origin and type of *Polygonatum cyrtonema* Hua.

### 2.4. FT-IR Spectra Characteristics of PCP 85−1−1

The FT-IR spectra of the polysaccharide sample are shown in Figure 4. The broad and intense absorption peak around 3310.69 cm^−1^ was attributed to the O–H stretching vibration [31], showing that there was a strong intermolecular force in the structure of the polysaccharides. The weak bands near 2934.14 cm^−1^ and 2884.30 cm^−1^ were assigned to the C–H stretching vibration. The absorption peak near 1643.89 cm^−1^ was the stretching vibration of C=O, and the absorption peak at 1414.08 cm^−1^ was the variable angle vibration of C–O, which is considered a characteristic absorption peak for plant polysaccharide structure [32]. These two absorption bands both belonged to the characteristic functional groups of polysaccharides. The two peaks at 1200 and 1000 cm^−1^ represented the stretching vibration of the C–O–C glycosidic bond in the furan ring [33]. The absorption peaks at 926.77 cm^−1^ and 871.40 cm^−1^ showed that PCP 85−1−1 contained a fructose ring skeleton containing β- glycosidic bonds [23].

### 2.5. Moisture-Proofing and Moisture Retention Experiment with Tobacco Shreds with PCP 85−1−1

#### 2.5.1. Moisture-Proofing Experiment

According to the moisture absorption and moisture retention characteristics of the polysaccharides in vitro, the ideal moisture retention/moisture-proofing humectants (PCP 85−1−1) for tobacco shreds were selected, and moisture-proofing tests were carried out. The moisture of the tobacco shreds was very sensitive to the ambient humidity [19]. The adsorption kinetic curves of the tobacco shreds with the addition of PCP 85−1−1, glycerol, and propylene glycol are shown in Figure 5A. The moisture content of the tobacco shreds increased rapidly in the first 4 h, gradually slowed down, and reached equilibrium in 10 h. The moisture content of the tobacco shreds in the control group was about 43.07 ± 1.16% ^d^, and the moisture content values of the tobacco shreds with 1% propylene glycol and glycerol were 38.38 ± 0.25% ^bc^ and 37.18 ± 0.68% ^ab^, respectively. The moisture content of the tobacco shreds with 1% PCP 85−1−1 was 36.03 ± 0.12% ^a^ (*^a–d^ indicates a significantly different from each other (analysis of variance, p < 0.05)*), and that with 2% PCP 85−1−1 was 35.55 ± 0.22%. When 1% PCP 85−1−1 was added to the tobacco shreds, the moisture resistance was higher than that of propylene glycol and glycerol (*p* < 0.05). These results showed that the addition of PCP 85−1−1 could slow down the moisture adsorption of tobacco shreds. Moreover, the moisture content values of the tobacco shreds with PCP 85−1−1 were all lower than those of control and blank, implying that PCP 85−1−1 was capable of improving the resistance of tobacco shreds to moisture. Ai et al. [34] also reported the same results and conclusions. They found that tobacco shreds with phosphorylated tobacco leaf polysaccharides had a certain moisture-proofing ability. However, phosphorylation of the tobacco leaf polysaccharides was required.

#### 2.5.2. Moisture Retention Experiment

The moisture retention curves of PCP 85−1−1 are shown in Figure 5B. The moisture content of the tobacco shreds decreased rapidly in the first 2 h, then gradually slowed down, and reached equilibrium in 8 h. The addition of traditional humectants, such as 1% glycerol and 1% propylene glycol, had a weak moisture retention effect on the tobacco shreds. The moisture retention of the tobacco shreds with 0.5% and 1% PCP 85−1−1 was lower compared to 1% glycerol and 1% propylene glycol. However, when 2% PCP 85−1−1 was added to the tobacco shreds, the moisture retention of the tobacco shreds was significantly improved and higher than the 1% glycerol and 1% propylene glycol (*p* < 0.05). Additionally, 2% PCP 85−1−1 had the highest moisture retention effect with an equilibrium moisture content of 12.12% (*p* < 0.05). Similar results were obtained in other reports. Huang et al. [35] concluded that the polysaccharide from *Wedelia prostrate* could effectively improve the moisture retention of tobacco shreds and reduce the moisture dissipation of tobacco shreds. However, the equilibrium moisture content of the polysaccharide from *Wedelia prostrate* was 9.28%, which was lower than the tobacco shreds with PCP 85−1−1. These results revealed that PCP 85−1−1 accelerated moisture adsorption and an increase in moisture adsorption capacity of the tobacco shreds by PCP 85−1−1, probably due to the hydrophilicity of PCP 85−1−1, which favors water adsorption through hydrogen bonds [2].

#### 2.5.3. Calculation of MRI and MPI

We used the Page model to analyze adsorption and desorption data, evaluate the half-life average moisture adsorption/desorption rate (v) as a kinetic indicator, and calculate MPI and MRI as a comprehensive index. The half-life average moisture adsorption/desorption rate represents the average value at which the change in moisture reaches the intermediate point (i.e., MR = 1/2). Tobacco shreds with low v values in both dry and humid environments are considered more stable. According to Equations (6) and (7), when M_e_ is the same, lower v correlates with higher MPI and MRI [13].

As shown in Table 2, the coefficient of determination (R^2^) of different tobacco shred samples for both desorption and adsorption processes were all above 0.950, and the mean relative percentage deviation modulus (E) ranged from 0.7181% to 1.6381%. E values lower than 10% suggest an adequate fit for practical applications [13]. Thus, the Page model gave a satisfactory fit to the experimental data and the model accuracy was acceptable to describe the adsorption and desorption processes of the tobacco shred samples under experimental conditions.

The Page model was used to further calculate the half-life average water adsorption/desorption rate v, MPI, and MRI. The results are shown in Table 3. In the process of moisture absorption, the tobacco shreds with 2% PCP 85−1−1 had the lowest moisture absorption rate (0.75 %/h), significantly smaller than with 1% glycerol and the control group. In terms of moisture absorption characteristics, PCP 85−1−1 added to the tobacco shreds was consistent with the moisture absorption performance of PCP 85 in vitro. In the process of moisturization, the water loss rate of the tobacco shreds mixed with 2% PCP 85−1−1 was the lowest at only 1.43 %/h, 0.07 %/h lower than 1% glycerol and 0.45 %/h lower than that of the control group (*p* < 0.05). Additionally, 2% PCP 85−1−1 had the maximum MRI and MPI values of 2.11 and 2.14, respectively. These results further explain that PCP 85−1−1 has a bilateral moisturizing ability. The improvement of the moisture retention capacity with the addition of PCP 85−1−1 could be explained by the hydrogen bonding effect at low RH. However, the improvement of the moisture-proofing ability could be explained by PCP 85−1−1 being a biological macromolecule which has certain film-forming properties. The film can block capillaries and weaken capillary adsorption [34,36].

### 2.6. Analysis of Taste and Aroma Components

#### 2.6.1. Electronic Tongue Evaluation

For consumers, the taste of cigarette smoke is very important. Therefore, we analyzed the taste change of the cigarette smoke after adding PCP 85−1−1 through electronic tongue technology. As shown in Figure 6, the response values of sweetness, bitterness, aftertaste-bitterness (aftertaste-B), and saltiness decreased in the artificial saliva after smoking tobacco shreds with humectants as compared to shreds without humectants. The results suggested that the addition of humectants may reduce the production of flavoring substances. The bitterness and aftertaste-bitterness (aftertaste-B) were significantly decreased with the humectants and they can improve the taste of the smoke (*p* < 0.05). However, the weakening of the sweetness was not conducive to the taste. Compared with the control group, the taste of the tobacco shreds with 2% PCP 85−1−1 was improved (e.g., for bitterness and aftertaste-B) and did not adversely affect the taste. This result was consistent with the previous reports. Ai et al. [34] found that the addition of phosphorylated tobacco leaf polysaccharides has a good effect on enhancing the sweetness and improving the fineness and softness of the smoke and can improve the smoking taste of cigarettes and reduce the dry feeling.

#### 2.6.2. Volatile Compound Contents

Tobacco shreds with different moisture contents had a different feeling from tobacco smoke. When the moisture content from the tobacco shreds was moderate, the feeling of the tobacco smoke was soft and comfortable, and the irritation was small. The combustion and cracking degree of the cigarette was executive, and the irritation and dryness of the smoke were strong under the low moisture content. However, when the moisture content was too high, the tobacco shreds had poor flammability, weak aroma, dull taste and lack of energy [37,38]. Zeng et al. [39] assessed the feeling of tobacco smoke in environments with different temperatures and humidity and found that with the increase in ambient humidity, the total content of flue gas components in both the gas phase and particulate phase decreased. The contents of aldehydes and ketones, organic acids, aromatic hydrocarbons, phenols, and nitrogenous compounds decreased, while the contents of higher fatty acids and their esters increased gradually. It can be speculated that the moisture content affected the change of volatile components in smoking.

The content of aroma components was calculated by the internal standard method. GC-MS data (Appendix A) showed that 62 volatile compounds were detected in the extract of tobacco shreds after smoking, including 7 nitrogenous compounds, 4 furans, 3 alcohols, 11 phenols, 12 ketones, 7 acids, 4 esters, and 14 hydrocarbons. The compounds were generated during the burning of the tobacco shreds and humectants.

The volatile compound content of the control group was 4651.61 μg/g, higher than the cigarettes with the addition of 1% glycerol (3607.35 μg/g) and the cigarettes with PCP 85−1−1 (4135.5 μg/g). However, the volatile compound content of cigarettes with PCP 85−1−1 was higher than the glycerol group. The majority of volatile components were nitrogenous compounds. Among them, nicotine had the highest content with a content of 1551.27–2472.44 μg/g. According to the research [40], a high content of nicotine is associated with a stronger aroma, taste, and strength of flue gas. However, excessive nicotine may lead to excessive irritation and the pungent taste of flue gas. Furans, major pyrolysis products from sugars and tobacco [41], are associated with cigarette aroma and are important aroma components of flue gas. The increase of furans, phenols, and ketones can enrich the fullness of the flue gas and improve the taste.

The content of different types of aroma compounds was analyzed by normalized z-score (Figure 7). The number of nitrogenous compounds, acids, and ester compounds decreased (*p* < 0.05), and furans, phenols, and ketones increased in the tobacco shreds with PCP 85−1−1 humectant compared to the group with glycerol. The increase of these substances may improve the taste and aroma of cigarettes when smoking. Similar results were obtained in other reports. Tobacco shreds with humectants produced a large number of pyrolysis products during smoking, mainly including ketones, aldehydes, esters, alkenes, acids, phenols, and other aroma substances that contribute to tobacco flavors, such as furfural, 5-methyl furfural, 2-methyl-1,3-cyclopentadiene, phenethyl phenylacetate, and benzene compounds [42]. It showed that the polysaccharides had the effect of enhancing the flavor while moisturizing the tobacco shreds by changing the moisture content of the tobacco shreds.

## 3. Materials and Methods

### 3.1. Materials and Chemicals

*Polygonatum cyrtonema* Hua was cultivated in Sanming City, Fujian Province, China. The dried rhizome of *Polygonatum cyrtonema* Hua tubers was ground into a fine powder using a high-speed disintegrator (DSY-9002, Yongkang jiu shun Ying Trading Co., Ltd., Jinhua, China). The powder was sieved through 80 mesh (Aperture 0.18 mm) and stored in a dryer for the extraction experiment. The tobacco shred samples (flue-cured tobacco) were obtained from China Tobacco Yunnan Industrial Co., Ltd. (Kunming, China). Glycerol (purity ≥ 99.0%), propylene glycol (purity ≥ 99.0%), ethanol (purity ≥ 99.7%), NaCl (purity ≥ 99.5%), dichloromethane (purity ≥ 99.5%), concentrated sulfuric acid (purity 95.0–98.0%), phenol (purity ≥ 99.0%), trifluoroacetic acid (purity ≥ 99.5%), Folin-phenol (1 mol/L), sodium nitrate (purity ≥ 99.0%), potassium bromide (purity ≥ 99.0%), and sodium dihydrogen phosphate (purity ≥ 99.0%) were purchased from Sinopharm Chemical Reagent Co., Ltd. (Shanghai, China). Gallic acid (purity ≥ 99.8%), Coomassie brilliant blue G-250 (purity ≥ 99.0%), bovine serum albumin (purity ≥ 99.0%) were purchased from Shanghai TITAN Technology Co., Ltd. (Shanghai, China). D-fructose (purity ≥ 99.0%), D-galactose (purity ≥ 99.0%), D-glucose (purity ≥ 99.0%), D-arabinose (purity ≥ 99.0%), L-fucose (purity ≥ 99.0%), L-rhamnose (purity ≥ 99.0%), D-mannose (purity ≥ 99.0%), D-xylose (purity ≥ 99.0%), and methanol (purity ≥ 99.9%) were from Sigma (St. Louis, MO USA). Hyaluronic acid was purchased from Yuanye Biotechnology Co., Ltd. (Shanghai, China).

### 3.2. Extraction and Fractionation Procedure

*Polygonatum cyrtonema* Hua powder was extracted at the optimal extraction temperature of 80 °C using the optimal material liquid ratio of 25 mL/g for 2 h, once. The extract was separated from *Polygonatum cyrtonema* Hua residue using gauze (Aperture 0.18 mm) and separated at 6000 r/min for 15 min by centrifugation (centrifuge 5415D, Eppendorf company, Saxony, Germany). After centrifugation, the supernatant was concentrated using a rotary vacuum evaporator (Re-52a, Shanghai Yarong biochemical instrument factory, Shanghai, China) at 50 °C.

PCP was obtained by fractional precipitation. Different volumes of absolute ethanol were added to the concentrated supernatant to create a series of ethanol concentrations (20%, 40%, 60%, and 85% *v*/*v*) successfully and incubated overnight at 4 °C. The precipitate was then collected by centrifugation at 4000 r/min for 20 min, freeze-dried, and four different fractions (PCP 20, PCP 40, PCP 60, and PCP 85) were obtained.

The content of polysaccharides was determined by the phenol sulfuric acid method with glucose as the standard [43], and the content of polyphenols was determined by Folin phenol colorimetry with gallic acid as the standard [44]. Protein content was measured using the Bradford test and bovine serum albumin (BSA) as the standard [45].

### 3.3. Moisture Absorption and Moisture Retention Properties of PCP

#### 3.3.1. Moisture Absorption Properties

The moisture absorption experiment was performed with modification as described before [46]. *Polygonatum cyrtonema* Hua polysaccharide samples were accurately weighed, along with sorbitol and hyaluronic acid in a small Petri dish and then placed in a desiccator containing saturated ammonium sulfate (relative humidity (RH) 81%) at 25 °C for 2, 4, 6, 8, 12, 24, and 48 h. The weight of each sample was accurately weighed, and the moisture absorption rate was calculated from the weight difference before and after moisture absorption. The moisture absorption was determined using Equation (1).
(1)$$\mathrm{Moisture}\;\mathrm{absorption}\;\left(\%\right)=\frac{m_{t}-m_{0}}{m_{0}}\times 100\%$$
where m_0_ was the weight of an oven-dried sample, and *m_t_* was the weight of the sample after moisture absorption for a specific time.

#### 3.3.2. Moisture Retention Properties

The weight of each sample was recorded at 48 h of the moisture absorption experiment, which was the first weight of sample moisture retention. The Petri dish was placed in a desiccator containing saturated sodium carbonate solution (relative humidity (RH) 43%) at 25 °C. The weight of each sample was accurately weighed and the moisture retention rate was calculated from the weight of the sample after water loss compared with the first weight of sample moisture retention [46]. The moisture retention was determined with Equation (2).
(2)$$\mathrm{Moisture}\;\mathrm{retention}\;\left(\%\right)=\frac{m_{t}}{m_{0}}\times 100\%$$
where m_0_ was the weight of the sample at the beginning of the moisture retention test and *m_t_* was the weight of the sample after moisture retention for a specific time.

### 3.4. Separation and Purification of PCP 85

For further purification, the crude polysaccharides (PCP 85) dissolved in deionized water were subjected to DEAE—Sepharose Fast-Flow column (2.6 cm × 40.0 cm). The column was eluted with a stepwise gradient with 200 mL of distilled water followed by 0.1, 0.2, and 1 M NaCl solutions at a 3 mL/min flow rate. The eluates (10 mL/tube) were collected, and the total carbohydrate contents were detected by using the phenol–sulfuric acid method. The water fraction was further purified with a Superdex 75 pg column (1.6 × 60 cm) using distilled water as eluent at a flow rate of 1 mL/min, and fractions were collected based on the peaks of eluting profile detected by the refractive index (RI) detector.

### 3.5. Monosaccharide Components Analysis of PCP 85−1−1

The analysis was based on the method of Wang [47] with appropriate modifications. PCP 85−1−1 was hydrolyzed at 110 °C for 4 h with 2 M trifluoroacetic acid (TFA, Sigma-Aldrich, St. Louis, MO, USA). The resulting samples were assessed using high-performance anion-exchange chromatography with a pulsed amperometric detection (HPAEC-PAD) for monosaccharide composition analysis and were detected using a PAD detector (Dionex, Sunnyvale, CA, USA) with a CarboPac™ PA20 column (Dionex, USA). The column was eluted with 2 mM NaOH and MilliQ water at a 0.45 mL/min flow rate.

### 3.6. Molecular Weight Analysis of PCP 85−1−1

Molecular weight analysis was performed with modification as previously described [47]. Briefly, the polysaccharide sample was dissolved in a 2 mL mobile phase and passed through the 0.45 μm microporous filter membrane. Pixel (7.8 mm) equipped with TSK GEL 3000PWXL (7.8 mm × 300 mm) and TSK-GEL G2500PWXL (7.8 mm × 300 mm) was then used in series with waters HPLC high-performance liquid chromatography (waters 2695, waters company, USA) to analyze and detect the molecular weight of the sample. The column was eluted with 0.15 mol/L NaNO_3_ and 0.05 mol/L NaH_2_PO_4_ (containing 0.02% sodium azide, pH = 7) with flow rate of 0.5 mL/min and column temperature of 30 °C, and the peak was detected with waters 2414 differential refraction detector and eight-angle laser light scattering detector.

### 3.7. FT-IR Spectroscopic Analysis of PCP 85−1−1

For FT-IR spectroscopy, 1 to 2 mg of the polysaccharide sample was mixed with 200 mg KBr, grounded, and pressed into a 1 mm KBr pellet. IR spectra were recorded on a Fourier-transform infrared (FTIR)-spectrometer, scanning between 4000 cm^−1^ and 500 cm^−1^.

### 3.8. Evaluation of Adsorption and Desorption Processes of Tobacco Shreds

#### 3.8.1. Moisture-Proofing and Moisture Retention of Tobacco Shreds with PCP 85−1−1

As described in Lin et al. [19], the tobacco shred samples were firstly preconditioned in a constant temperature and humidity box (JYK-253, JIAYU Scientific instrument, Shanghai, China) at 22 °C and RH 60% for 48 h. PCP 85−1−1 as humectants were selected from 3.4 for the tobacco shreds moisture absorption and retention experiment. Propylene glycol and glycerol were used as positive controls, and the control was added with water in the same amount. Tobacco shreds in each group were evenly sprayed with the humectant solution with three different mass fractions (0.5%, 1%, and 2%), stirred, and divided into three parallel groups. The weight of the Petri dish was recorded in a dry state and the tobacco shreds were put into the Petri dish. The tobacco shreds with humectants were stored at 22 °C and RH 60% for a minimum of 72 h. The weight of the balanced tobacco shreds was recorded.

The balanced tobacco shreds were placed at 22 °C/RH 85% and 22 °C/RH 33% for adsorption and desorption experiments. The moisture content of tobacco shreds was calculated according to the difference between the quality of tobacco shreds before and after water absorption and the dry base weight of tobacco shreds. The final moisture content was considered the equilibrium moisture content (EMC). The initial moisture content of tobacco shreds was determined using the oven-drying method at 100 °C for 2 h [48]. The moisture content of the sample at a special time was calculated based on the initial moisture content and the change of weight over time.

#### 3.8.2. Calculation of MRI and MPI

MRI (moisture retention index) was calculated as described by Lou et al. [49] to evaluate the moisture retention capacity of tobacco shred samples, and MPI (moisture-proofing index) was introduced in this paper to evaluate the moisture-proofing capacity. Firstly, the data for the adsorption and desorption processes of tobacco shred samples were fitted to the Page model:(3)$$\mathrm{MR}=\frac{M_{t}-M_{e}}{M_{0}-M_{e}}=\exp\left(-kt^{n}\right)$$
where MR is the moisture ratio; M_0_ is the initial moisture content, (%); M_e_ is the equilibrium moisture content, (%); *M_t_* is the moisture content at a given time, (%); *t* is the adsorption or desorption time, (h); and n and k are model constants. The fitting parameters were estimated using non-linear regression analysis with Origin 9.0 and the fit was evaluated using R^2^ and E.
(4)$$E=\frac{100}{n}\overset{n}{\underset{i=1}{\sum }}\left|\frac{M_{i,\exp}-M_{i,\mathrm{pre}}}{M_{i,\exp}}\right|$$
where M_i,exp_, and M_i,pre_ are experimentally observed and model-predicted moisture contents, respectively; n is the number of data points.

The half-life average moisture adsorption/desorption rate v (%/h), MPI, and MRI were calculated with Equations (5)–(7), respectively:(5)$$v=\frac{|M_{0}-M_{e}|}{t_{1/2}}$$
(6)$$\mathrm{MRI}=2-\frac{v}{\bar{v}}+\frac{M_{e}}{\bar{M_{e}}}$$
(7)$$\mathrm{MPI}=4-\frac{v}{\bar{v}}+\frac{M_{e}}{\bar{M_{e}}}$$
where t_1/2_ represents the time when MR = 1/2 (h); $\bar{v}$ represents the average value of v for different samples, (%/h); ${\bar{M}}_{e}$ represents the average value of M_e_ for different samples, (%).

### 3.9. Aroma Analysis

#### 3.9.1. Detection and Analysis by Electronic Tongue

As an instrument for rapid detection of food flavor, the electronic tongue can solve the shortcomings of artificial sensory evaluation, such as time and high cost, and can objectively evaluate food flavor [50]. The tobacco shreds mixed with PCP 85−1−1 were used for the electronic tongue and GC-MS analysis. Tobacco shreds sprayed with humectants (2% PCP 85−1−1, 1% glycerol and the same amount of distilled water), prepared as described in 3.8, were made into cigarettes and stored at 22 °C and 60% RH for 48 h.

According to the GB/T 19609-2004 in China [51], cigarettes with different humectants were burned and sucked, and the fume was intercepted by a Cambridge filter. Each filter intercepted 20 cigarettes. Artificial saliva was prepared by dissolving 2.2365 g KCl and 0.045 tartaric acids in 1000 mL deionized water.

The smoke was collected in artificial saliva, with a constant volume of 200 mL. Sensory analysis of artificial saliva was performed with an electronic tongue, and three parallel tests were performed for each group.

#### 3.9.2. GC-MS Analysis

Consistent with the smoking method in Section 3.9.1, we changed the artificial saliva to dichloromethane to collect the flue gas. After smoking, the Cambridge filter and dichloromethane were transferred into a conical flask. Aroma components were extracted at 10 °C for 3 h, and the volume of the extract was maintained constant at 100 mL. The internal standard heptadecane was added to the extraction solution, blown with nitrogen, and dried with anhydrous magnesium sulfate. The extract was filtered through a 0.45μm microporous organic filter membrane for GC-MS.

Chromatographic analysis was performed using a QP2010 series GC–MS (Agilent Corporation, Palo Alto, CA, USA) equipped with an Agilent 19091S column (60 m × 0.25 mm × 0.25 μm). The injection volume was 0.2 μL. The injection temperature was 250 °C. All samples were injected in split mode with a split ratio of 5:1 and a flow rate of 1.0 mL/min. The initial temperature was maintained at 50 °C for 4 min and then raised to 300 °C at a rate of 4 °C/min and maintained for 10 min. The solvent delay time was 6 min. Mass spectra were acquired in full-scan mode with repetitive scanning from 30 *m*/*z* to 300 *m*/*z* in 1 s. The mass spectrum transmission line temperature was 280 °C. The ion source temperature was 230 °C. The electronic ionization (EI) source energy was 70 eV, and the quadrupole temperature was 150 °C.

### 3.10. Statistical Analysis

Results are presented as mean ± standard error of the mean (SEM). The statistical significance was analyzed by student’s *t* test using SPSS software (SPASS, version 23.0 SPSS Inc., Chicago, IL, USA) for mean differences among the samples by one-way analysis of variance (ANOVA). Differences between groups at *p* < 0.05 were considered statistically significant.

## 4. Conclusions

In conclusion, *Polygonatum cyrtonema Hua* polysaccharide (PCP 85−1−1) was obtained by fractional alcohol precipitation, and its structural composition, moisture absorption, and moisturizing characteristics were studied. PCP 85−1−1 was classified as homogeneous polysaccharide by the molecular weight analysis and the monosaccharide composition analysis and mainly consisted of fucose, glucose and fructose, and the molar ratio was 22.73:33.63:43.65. The FT-IR spectra characteristics of PCP 85−1−1 showed that PCP 85−1−1 contained a furan ring skeleton with β-glycosidic bonds. Moisture absorption and the moisturization test showed that PCP 60 and PCP 85 had a good two-way moisturizing effect. The ability of PCP 85−1−1 to improve the moisture retention and moisture-proofing characteristics of porous carbohydrates was analyzed. The result revealed accelerated moisture adsorption and an increase in moisture adsorption capacity of the tobacco shreds with PCP 85−1−1 under dry conditions, and PCP 85−1−1 can slow down the moisture adsorption in tobacco shreds under high relative humidity. It was concluded that PCP 85−1−1 had excellent moisture retention and moisture-proofing characteristics. Finally, the aroma analysis showed that the addition of PCP 85−1−1 improved the increase in furans, phenols, and ketones and can enrich the fullness of flue gas and improve the taste by changing the moisture content. In conclusion, *Polygonatum cyrtonema* Hua polysaccharides can be utilized as a natural humectant on porous carbohydrates (tobacco shreds).

## Figures and Tables

**Figure 1 molecules-27-05015-f001:**
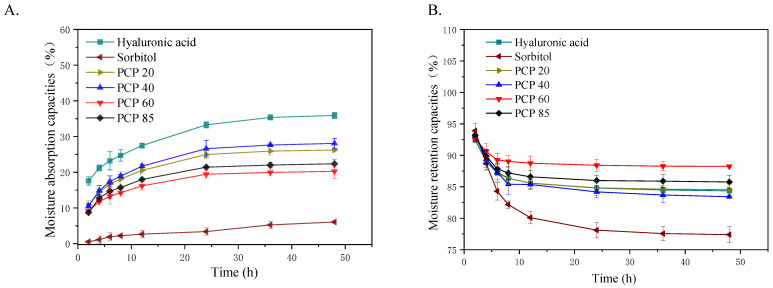
(**A**) The moisture absorption of samples at RH = 81%. (**B**)The moisture retention of samples at RH = 43%.

**Figure 2 molecules-27-05015-f002:**
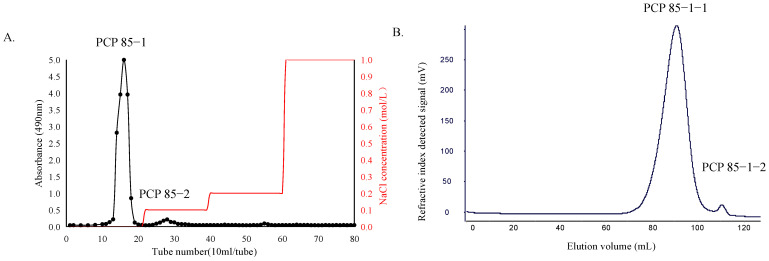
(**A**) Stepwise elution curve of crude PCP 85 on DEAE-Sepharose Fast Flow cellulose column. (**B**) Stepwise elution curve of PCP−85−1 on Superdex 75 pg column.

**Figure 3 molecules-27-05015-f003:**
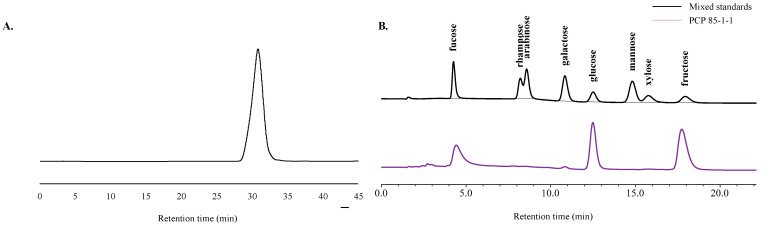
(**A**) The HPGPC-RID profiles of PCP 85−1−1; (**B**) HPLC-RID chromatograms of standard monosaccharide mixture and hydrolysis products of PCP 85−1−1.

**Figure 4 molecules-27-05015-f004:**
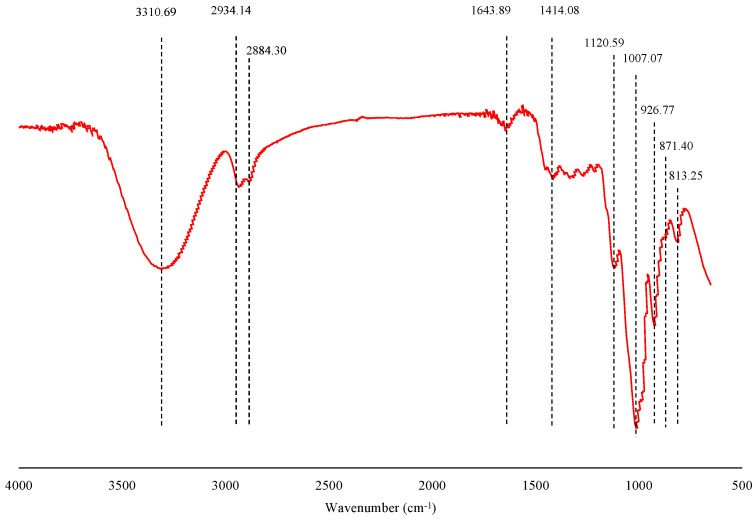
FT-IR spectra of the PCP 85−1−1.

**Figure 5 molecules-27-05015-f005:**
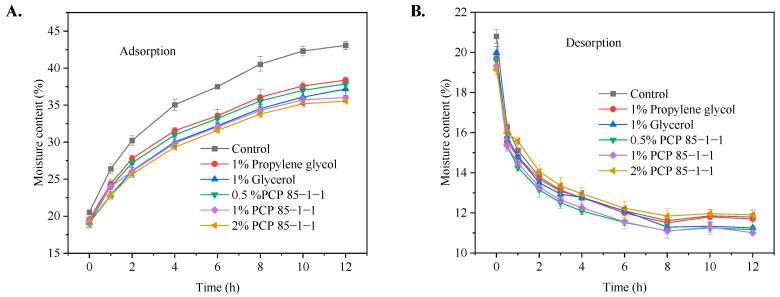
(**A**) Moisture adsorption of tobacco shreds with PCP 85−1−1. (**B**) Moisture desorption characteristics of tobacco shreds with PCP 85−1−1.

**Figure 6 molecules-27-05015-f006:**
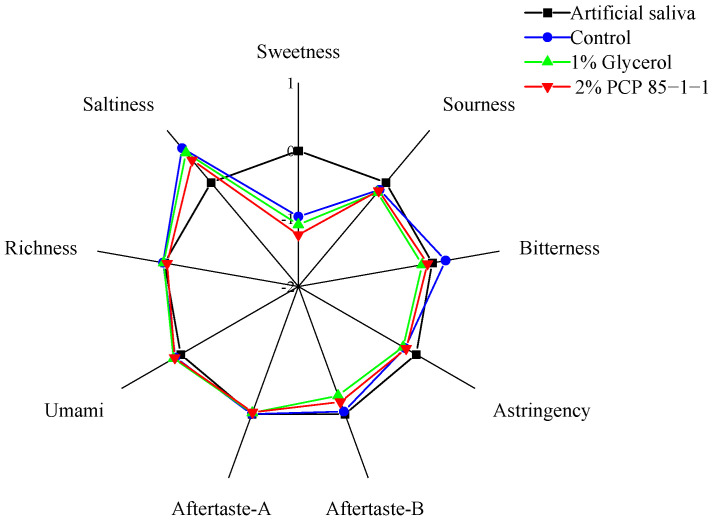
Radar chart of electronic tongue data for the taste of tobacco shreds with humectants.

**Figure 7 molecules-27-05015-f007:**
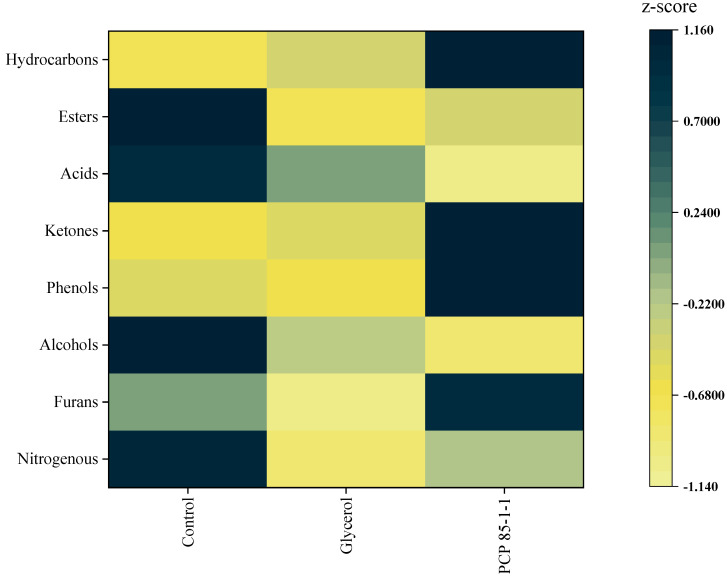
Hot map of the kinds of volatile compounds in the extract of tobacco shreds after smoking.

**Table 1 molecules-27-05015-t001:** The chemical compositions of *Polygonatum cyrtonema* Hua polysaccharides.

	PCP 20	PCP 40	PCP 60	PCP 85
Polysaccharide yield (%)	1.34 ± 0.88 ^a^	5.32 ± 0.54 ^c^	2.47 ± 0.61 ^b^	15.43 ± 1.21 ^d^
Purity (%)	62.44 ± 1.64 ^d^	56.64 ± 3.57 ^a^	57.77 ± 1.58 ^b^	58.18 ± 2.25 ^c^
Protein (mg/g)	3.98 ± 0.0075 ^c^	3.15 ± 0.10 ^a^	3.83 ± 0.015 ^c^	3.45 ± 0.11 ^b^
Polyphenol (mg/g)	5.54 ± 0.21 ^d^	2.76 ± 0.059 ^c^	2.28 ± 0.036 ^b^	1.61 ± 0.001 ^a^

^a–d^ Values within a row followed by a different lowercase letter are significantly different from each other (analysis of variance, *p* < 0.05).

**Table 2 molecules-27-05015-t002:** Estimated Page model parameters and fitting criteria of tobacco shred samples with different humectants in adsorption and desorption processes.

Sample	Adsorption				Desorption			
	k	*n*	R^2^	E (%)	k	*n*	R^2^	E (%)
Control	0.7884	0.4148	0.9939	0.9367	0.2727	0.8418	0.9780	1.5053
1% Propylene glycol	0.7497	0.4089	0.9935	0.9820	0.2791	0.8206	0.9794	1.5015
1% Glycerol	0.7179	0.4089	0.9947	1.0144	0.2220	0.8876	0.9804	1.6381
0.5% PCP 85−1−1	0.7860	0.3947	0.9968	0.9496	0.2809	0.8119	0.9796	1.4992
1% PCP 85−1−1	0.7061	0.4361	0.9933	1.0179	0.2346	0.8668	0.9770	1.6104
2% PCP 85−1−1	0.6694	0.4208	0.9932	1.0789	0.2290	0.8791	0.9800	1.6179

**Table 3 molecules-27-05015-t003:** EMC, v, MPI, and MRI of tobacco shred samples with different humectants in adsorption and desorption processes.

Sample	Adsorption			Desorption		
	EMC (%)	v (%/h)	MRI	EMC (%)	v (%/h)	MPI
Control	43.07	0.88	1.95	11.79	1.88	1.67
1% Propylene glycol	38.38	0.80	2.05	11.69	1.57	2.00
1% Glycerol	37.18	0.84	1.98	11.25	1.50	2.07
0.5% PCP 85−1−1	37.86	0.83	1.95	11.16	1.56	2.01
1% PCP 85−1−1	36.03	0.82	1.96	11.01	1.45	2.11
2% PCP 85−1−1	35.55	0.75	2.11	12.12	1.43	2.14

## Data Availability

The presented data are available upon reasonable request from the corresponding author.

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
