# Peer review of "Structural Characterization of a Polygonatum cyrtonema Hua Tuber Polysaccharide and Its Contribution to Moisture Retention and Moisture-Proofing of Porous Carbohydrate Material"

_molecules, 2022, doi:10.3390/molecules27155015_

Round 1
Reviewer 1 Report
Reviewer Comments:
Highlights: Provided and helpful. However, please check GFA for the word limit. The highlights have exceeded 85 characters including space.
Graphical Abstract:
Provided and helpful
General comment:
This paper presented a report of moisture retention and moisture-proof functions of Polygonatum cyrtonema Hua polysaccharides. Specifically the effect of Polygonatum cyrtonema Hua polysaccharides on adsorption and desorption properties of tobacco shreds were investigated. With regard of effect of the polysaccharides on changes in taste and aroma, the samples were analyzed using electronic tongue and gas chromatography mass spectrometry. The result is promising, indicating that the polysaccharides could increase both moisture-retention and moisture-proof of the tobacco. Overall, a great amount of effort has been put into the experimental design and analysis. However, there are still some mistakes and issues in the structure of the manuscript. Therefore, I recommend a major revision. Hope the comments below help.
Specific comment:
Abstract:
- The background of the study is provided but the knowledge gap in the study seems to be missing.
- Also, significance of the study and how the findings can be used to advance the field should be included.
- An abstract is often presented separately from the article, so it must be able to stand alone.
- Please try to merge all information into a paragraph with some attractive and new findings. The main result from the review is not seem in the abstract.
- Kindly refer some latest papers as the structure of the paper is useful to this report. Example, biological remediation of acid mine drainage: Review of past trends and current outlook; microalgae biorefinery: High value products perspectives; recent advances in downstream processing of microalgae lipid recovery for biofuel production
Introduction:
- First line: does the author mean “genus”?
- Line 42: split up the long sentence for better understanding.
- There is no continuity from the sentence explaining about purification of polysaccharides to the next sentence about types of polysaccharides. Kindly improve the “flow”.
- Line 66-70: Provide citation.
- Similarly, provide literature about the current state of issue related to moisture-proof of porous carbohydrates, then highlight the existing bottlenecks, leading to explanation about the knowledge gap.
- Revised Introduction section based on the structure below:
1st paragraph: Problem statement
2nd paragraph: Current ongoing solution
3rd paragraph: Proposed solution in this work.
4th paragraph: Summarized the current research novelty and objective of this work.
- There are some tips that improve structure from this paper that authors are recommended to refer: “Incorporating biowaste into circular bioeconomy/ A critical review of current trend and scaling up feasibility”.
- Problem statement of your introduction is not strong, need to discuss more about it.
- The earlier paragraphs should lead logically to specific objectives of the study.
Results and discussion:
- The major problem in this section is that most findings are just presented without critical evaluation and appropriate redaction.
- More discussion and citations are needed to warrant a publication.
- Compare the findings of current study with similar studies in the literature.
- Provide underlying mechanism or chemical properties of the PCP ascribing to disparity in moisture-retention and moisture-proof properties.
- Discussion on the FTIR spectra needs to be improved. Explain the shift of peak or lack thereof.
-
Materials and methods:
- Line 261: Provide size of the mesh
- Provide some characterization data on the sample.
- Provide country of origin for the chemicals.
- Similarly, provide the pore size of the gauze.
References
- There are insufficient references in text especially in results and discussion. More citation is needed to benchmark the findings of current study to that of the literature.
Reviewer 2 Report
I want to thank the authors for this good and interesting work. Wish you all the best. However, the paper could be acceptable after addressing the following comments-
1. The abstract is well written.
2. Highlights keyword could be substituted by other keywords as the title contains the same word.
3. In sentence 32 the word “genius” should be checked for the right spelling.
4. In Figure 1, the authors have studied the moisture absorption at 81%RH and moisture retention at 41% RH. What is the basis of this definite RH?
5. Line 241 should be written again for its clarification.
6. The results and discussions of the study need to be more supportive and elaborative. The authors represented the results but the discussion needs to be included.
7. In the result part lines 250-252 “The number of nitrogenous compounds, acids, and esters compounds decreased (p<0.05), and furans, phenols, and ketones increased in the tobacco shreds with PCP 85-1-1 humectant, as compared to the group with glycerol.” The authors must include the mechanism and literature support for the increase/decrease of the chemical compounds after adding PCP 85-1-1.
8. In section 3.1. Materials and chemicals, purity of the chemicals should be added.
9. The authors made a fractionization of 20%, 40%, 60%, and 85%, v/v, Why author didn’t check for 80% and more than 85%? Because PCP 85% has been claimed as the best one by the authors. I will suggest reporting 80% and 90% fractions.
10. The conclusions need to be more specific and improved. The abstract and conclusion authors wrote contain a very small difference.
Reviewer 3 Report
This is a study of polysaccharides from Polygonatum cyrtonema Hua and the characterisation of one such complex molecule designated PCP 85-1-1. The work is well presented and seems to have been executed correctly. Why not state that the compound PCP 85-1-1 is a tuber-derived polysaccharide? Is PCP 85-1-1 found in quantities in other organs of the plant? If so, do these have the same humectant-like properties? Aside from some minor changes required, I am of the opinion that this manuscript meets all the requirements for publication in the dpi journal molecules.
Aspects that require attention include;
Pg 1, ln 1-3 suggest revision of title to read "Structural characterization of a Polygonatum cyrtonema Hua tuber polysaccharide and its contribution to moisture-retention and moisture proofing of porous carbohydrate material"
Pg 1, ln 16 suggest that 'dampness' (which is the result of moisture uptake/deposition/condensation) be changed to read 'become damp'
Pg 2, ln 49 & 52 Change "Polygonatum Cyrtonema" to read 'Polygonatum cyrtonema' i.e. lower case for species
Pg 2, ln 82 change "was' to 'is'
Pg 3 96 & change "was' to 'is'
ln 255 in Figure 7 legend, change "hot chart" to read 'heat map'
A better characterisation of the carbohydrates from the studied species using UHPLC-MS coupled with NMR might have elevated this manuscript and allowed for a more complete description of the molecular structure of PCP 85-1-1.
Round 2
Reviewer 1 Report
The authors have addressed all the comments adequately.
Reviewer 2 Report
Thank you for the remarkable development of the manuscript.